# BED: Boundary-Enhanced Decoder for Chinese Word Segmentation

## Abstract

Chinese Word Segmentation (CWS) is an essential fundamental step in the Chinese NLP processing pipeline. In recent years, with the development of deep learning and pre-training language models, multitudinous CWS models based on pre-training models have been proposed, and the performance of CWS models has been dramatically improved. However, CWS remains an open problem that deserves further study, such as CWS models are subjected to OOV words. To our knowledge, currently proposed CWS approaches mainly refine on the encoder part of CWS model, such as incorporating more word information into the encoder or doing pre-training related to the CWS task, etc. And there is no attempt to improve the decoder's performance of CWS model. This paper proposes an optimized decoder for the CWS model called Boundary-Enhanced Decoder (BED). It could bring 0.05% and 0.69% improvement on Average-F1 and OOV Average-F1 on four benchmark datasets. We also publish our implementation of BED.

## 1 Introduction

Chinese Word Segmentation (CWS) is an essential step in the Chinese NLP processing pipeline. In languages like Chinese, there are no evident word boundaries in a sentence. Unlike English, sentences are naturally split into separate words by spaces in the text. When we do a Chinese NLP task, we are required to segment the sentence into words at first and then feed words into a downstream model. Although character-level models have achieved good results on many NLP tasks in recent years, many studies have shown that models incorporating word information can improve their performance Tian et al. (2020); Liu et al. (2021); Zhang & Yang (2018). So a good word segmentation model is significant for Chinese NLP tasks.

Although the effect of the Chinese word segmentation model has been dramatically improved with the recent development of the pre-training model, there are still multitudinous problems in CWS. For example, the model usually performs poorly on difficult segmentation words, and the OOV words are one of them. OOV words are words for which the model was not taught how to segment them out in the training phase. Many studies show that the model often has a poorer segmentation effect for OOV words than common words. To alleviate and solve this problem to a certain extent, we propose the boundary-enhanced decoder (BED) in this paper.

Our proposed method is inspired by the process of humans doing word segmentation tasks. The difficulty of word segmentation is different for each word in one sentence. For some words in a sentence, it is easy for us to figure out how to segment them, while others need repeated scrutiny and pondering to be correctly segmented. Therefore, when people perform word segmentation, they tend to preliminarily segment easy and relatively specific words' boundary, such as punctuation and some transition words, so that the sentence is roughly divided into large blocks. Then, the content in large blocks is more finely segmented.

For example, when we segment the sentence in Figure 1, "王钟翰/论证/后金/在/进入/辽沈/前/，/已/属/奴隶制/社会/。(Wang Zhonghan argues that Post-Jin had already belonged to a slave society before entering Liao and Shen.)". We can split it into coarse-grained parts effortlessly, like the second layer in Figure 1. Blocks with a red background are words. All words can be determined in the first segmentation, except for the "论证后金 (argues Post-Jin)" part with yellow background needs to be more fine-grained segmentated. There are two ways to segment it, one way is "{论证, 后, 金} ({argues, after, Jin})" and another is "{论证, 后金}({argues, Post-Jin})" which

Figure 1: A human CWS case.

is the correct segmentation. It is demanding for us to segment these words correctly. However, if we segment the relatively easy blocks of the sentence into words and then the hard block could be easy to be segmented, because of the excluding of other information's interference.

Therefore, we propose BED, which imitates the process mentioned above. The model looks for easily tokenized positions, divides a whole sentence into several parts first, and then tokenizes each piece into more fine-grained words. This more intuitive approach can further improve the performance of the word segmentation model, especially the performance of OOV words. To our best knowledge, it is the first study trying to optimize the decoder part in a CWS model.

In this paper, we will first introduce the related work of Chinese word segmentation in section 2 . Then we define the problem of CWS and introduce our proposed decoder and the entire model structure. We conduct exhaustive experiments, and the results and analysis will be described in Section 5.

## 2 RELATED WORK

Recently, with the development of pre-training models, many different pre-training models have been proposed Devlin et al. (2018); Liu et al. (2019); Clark et al. (2020); Zhang et al. (2019); Floridi & Chiriatti (2020); Lan et al. (2019); Bai et al. (2021); Dong et al. (2019). Approaches based on the pre-training model have achieved SOAT in many tasks of NLP Rajpurkar et al. (2016); Wang et al. (2018). The same is true for the CWS task. Some recently proposed methods based on pre-training models have greatly improved the model's effectiveness on the CWS. These methods exploit a pre-trained masked language model as the text encoder and a crf layer or softmax layer as the decoder Liu et al. (2021); Ke et al. (2020); Huang et al. (2019); Tian et al. (2020); Meng et al. (2019); Huang et al. (2021). And then, the model is finetuned on the CWS-related corpus.

Among these studies, some try to leverage external information, such as vocabularies, from the external corpus. Liu et al. (2021); Tian et al. (2020) both attempt to incorporate word information into the encoder. They all use character-level pre-trained models. During finetuning, the embeddings of the words in the dictionary contained in the sentence are added to the corresponding character position in the sentence. According to Liu et al. (2021) rule, different locations of fusing word information, to categorize models incorporating word information, Tian et al. (2020) fuses the word representation at the model level and Liu et al. (2021) fuses at the BERT level. These two fusion models improve the CWS model performance compared to models only using character-level information.

Some works use one model to handle CWS datasets with distinct tokenization criteria uniformly. Ke et al. (2020) proposed a pre-training model specifically for CWS. The pre-training model which has a specific input position representing the word segmentation standard, makes it can unify different CWS criteria. And to be able to adapt to the new segmentation criteria in the fine-tuning stage, they used a meta-learning algorithm in the pre-training stage. Huang et al. (2019) proposed a model that uses BERT as the text encoder, and it extracts two kinds of information: one is the common information which shared by all criterions and another is the information only belonging to each

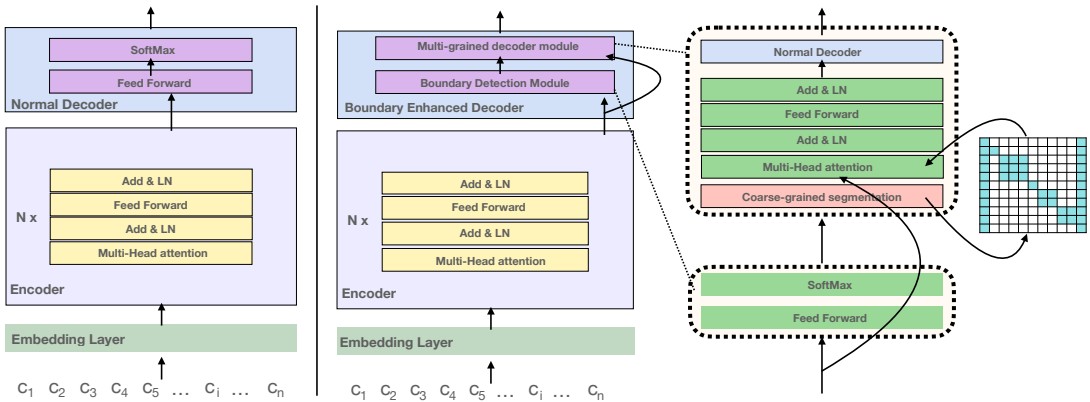

Figure 2: The architecture of BED-CWSM. The left one is a CWS model with a normal decoder. And the right one is BED-CWSM, which has a Boundary-Enhanced Decoder (BED). There are two modules in BED: Boundary Detection Module and Fine-grained Decoder Module.

criteria, from the output of the encoder in the upper layer. And then it fuses these two kinds of information to improve the model performance on multi-criteria CWS data. At the same time, some works try to apply multimodality to CWS. For example, Meng et al. (2019) try to use the CNN model to extract the image features of characters in sentences so that the CWS model can impose the graphic information of characters.

However, all the above optimizations are related to the encoder of CWS models. To our knowledge, no published research has optimized the decoder for the CWS model. In this paper, we propose a novel CWS decoder structure inspired by the process of humans doing CWS. It can further improve the performance of the CWS model.

## 3 PROBLEM

In CWS task, we are given a sentence $S = \{c_1, c_2, ..., c_i, ..., c_n\}$, which contains $n$ characters, $c_i$ is the i-th characters in the sentence. We are required to tokenize $n$ characters into $m$ words, and each word contains one or more chars. So we can denote a tokenized sentence in word granularity $S = \{w_1, w_2, ..., w_k, ..., w_m\}$, $w_k$ is the k-th word in the sentence and $w_k = \{c_{w_{k1}}, ..., c_{w_{kt}}\}$, there are $t$ chars in $w_k$.

We can treat the CWS task as a sequence label task whose output space is $\Upsilon = \{B, M, E, S\}$. To tokenize chars into words, normally, a CWS model needs to label each token into one of four symbols in $\Upsilon$. Chars labeled with $B$ mean they are the beginning of words. Chars marked as $M$ are the middle elements of words. Label $E$ represents a char is the end of a word, and $S$ means an individual char is a word itself. Then we can split a sentence into words according to the label of each char.

## 4 APPROACH

The whole architecture of our proposed model, BED-CWSM, is shown in Figure 2. Compared with a CWS model with a normal softmax decoder, our model has the same encoder structure and different decoder modules. BED-CWSM has a Boundary-Enhanced Decoder (BED), and there are two modules in BED: Boundary Detection Module and Fine-grained Decoder Module. The encoder module of both models is based on char-level pre-trained language model (Pr-LM) such as BERT Devlin et al. (2018) and Roberta Liu et al. (2019).

### 4.1 EMBEDDING LAYER

The embedding layer is the same as the embedding layer in BERT. It contains two embedding matrixes: token embedding matrix $M_{et}$ and positional embedding matrix $M_{ep}$. The shape of $M_{et}$ is

$V * d$, and $M_{ep}$ is $L_m * d$. $V$ is the vocabulary size of the model, and $L_m$ is the limited max length of the input sentence. For an input char $c_i$, its embedding is computed as equation 1. $id$ is a function to get the index of $c_i$ in the model's vocabulary. For a sentence has n chars, the embedding layer outputs all chars' embedding $E = \{e_1, e_2, ..., e_i, ..., e_n\}$. The first char in all sentences is a special token $[CLS]$, and the last char is a special token $[SEP]$.

$$e_i = M_{et}[id(c_i), :] + M_{ep}[i, :] \tag{1}$$

## 4.2 ENCODER

The encoder comprises N transformer encoder layers, and each transformer encoder layer takes the previous layer's output as input. We denote the output of $l$-th transformer encoder layer as $H^l$. For each transformer encoder layer, we compute as following equations:

$$\begin{aligned} A &= LN(H^{l-1} + MHA(H^{l-1}) \\ H^l &= LN(A + FFN(A)) \end{aligned} \tag{2}$$

In equation 2, LN represents layer normalization, MHA denotes multi-head attention operation and FFN means feed-forward network with ReLU activation function. $H^0$ is equal to $E$. We denote the output of final transformer encoder layer as $H^L = \{h_1^L, h_2^L, ..., h_i^L, ..., h_n^L\}$, $h_i^L$ is the final representation of i-th char in the sentence.

## 4.3 NORMAL DECODER

Two kinds of decoders always are adopted in the CWS model, softmax and crf. They all take $H^L$ as input and compute each char's probability of output space $\{B, M, E, S\}$. Softmax decoder is simple and computation efficient. A crf decoder models the dependency between successive labels. However, there is no conclusion that a model with a softmax decoder is worse than a crf decoder, especially for a model with a Pr-LM encoder. So we compare both decoders in this paper.

### 4.3.1 SOFTMAX DECODER

A softmax decoder is simple. For i-th characters, we calculate its label probabilities as equation 3. $FFN$ is a feed-forward network, and $softmax$ represents a softmax operation. We use $P$ to denote predicted probabilities of all chars in a sentence, which $P = \{p^1, p^2, ...p^i, ...p^n\}$.

$$p^i = softmax(FFN(h_i^L)) \tag{3}$$

For the softmax decoder, the CrossEntropy loss of $m$ samples is computed as follows:

$$L_D = -\frac{1}{m}\sum_j^m \frac{1}{L_j}\sum_i^{L_j} y_j^i log(p_j^i) \tag{4}$$

$y_j^i$ is the label of i-th char in the j-th sample, and $p_j^i$ is the probability model predicted for i-th char in the j-th sample. $L_j$ is the length of the j-th sample.

### 4.3.2 CRF DECODER

In a crf decoder, it calculates logits of each char $O$ with a $FFN$ layer firstly.

$$\begin{aligned} o^i &= FFN(h_i) \\ O &= \{o^1, .., o^i, ...o^n\} \end{aligned} \tag{5}$$

For a given label sequence $y = \{y_1, y_2, y_3, ..., y_i, ..., y_n\}$, the probability of this sequence is computed as follow:

$$P = \frac{exp(\sum_i (O_{i,y_i} + T_{y_{i-1},y_i}))}{\sum_{\hat{y}}(exp(\sum_i (O_{i,y_i} + T_{y_{i-1},y_i})))} \tag{6}$$

$\hat{y}$ in equation 6 is all possible label sequences and $T$ is the transition score matrix. The loss of the crf decoder is calculated as follows:

$$L_D = -\frac{1}{m}\sum_j^m log(P) \tag{7}$$

In the inference phase, we get the label sequence achieving the highest $P$ using the Viterbi algorithm.

### 4.4 BOUNDARY-ENHANCED DECODER

Boundary-enhanced decoder has two modules, the boundary detection module and the multi-grained decoder module, as shown in figure 2.

#### 4.4.1 BOUNDARY DETECTION MODULE

The boundary detection module can determine whether a char is at an end position of a word. It takes $H^L$ as input and outputs binary classification probability $\tilde{P}$ of all chars.

For a segmentation label at i-th position of j-th sample $y_j^i$, we convert it to binary classification boundary detection label $\tilde{y}_j^i$ as equation 8. We give all chars that are the ending of a word or themself are a word a positive label 1 and all other chars a negative label 0.

$$\tilde{y}_j^i = \begin{cases} 1, & y_j^i \in \{E, S\}; \\ 0, & y_j^i \in \{M, B\}. \end{cases} \tag{8}$$

Boundary probability $\tilde{P}$ computed by equation 9. It simply passes $H^L$ into a feed-forward network and converts its output into a probability with a softmax operation.

$$\tilde{P} = softmax(FFN(H^L)) \tag{9}$$

The boundary detection loss $L_{BD}$ is computed as follow:

$$L_{BD} = -\frac{1}{m}\sum_j^m \frac{1}{L_j}\sum_i^{L_j} \tilde{y}_j^i log(\tilde{p}_j^i) \tag{10}$$

#### 4.4.2 MULTI-GRAINED DECODER MODULE

Multi-grained decoder module takes $H^L$ and $\tilde{P}$ as inputs. This module contains a coarse-grained segmentation part, a transformer encoder layer, and a normal decoder layer.

The input of coarse-grained segmentation part is $\tilde{P}$, and it will split the sentence into several segment according to $\tilde{P}$. It will output a matrix $M_{seg}$ which can represent corse-grained segmentation result of a sentence. And this matrix will be passed into subsequent transformer encoder layer as its attention mask matrix. When $j \in \{0, L\}$, $M_{seg}[i, j] = 0$, and when $j \notin \{0, L\}$, it is computed as equation 11.

$$M_{seg}[i,j] = \begin{cases} -inf, & max(\tilde{p}(i:j)) \geq thres_{seg} \\ 0, & others \end{cases} \tag{11}$$

$L$ is the length of the input sentence. The first and last character in the input sentence is $[CLS]$ and $[seq]$ token, respectively. $thres_{seg}$ is a threshold we need to set. The larger value of $thres_{seg}$, the more confident coarse-grained segmentation the model will do and fewer segments in a sentence.

|          | MSR | | PKU | | AS | | CITYU | |
|----------|-------|------|-------|------|-------|------|-------|------|
|          | Train | Test | Train | Test | Train | Test | Train | Test |
| CHAR     | 4,050K | 184K | 1,826K | 173K | 8,368K | 198K | 2,403K | 68K |
| WORD     | 2,368K | 107K | 1,110K | 104K | 5,500K | 123K | 1,456K | 41K |
| OOV RATE | -     | 2.7  | -     | 5.8  | -     | 4.3  | -     | 7.2  |

Table 1: Statics of four benchmark datasets, in terms of the number of characters and words. OOV RATE is the percentage of out-of-vocabulary (OOV) words in test set.

When we set $thres_{seg}$ larger than 1, $M_{seg}$ becomes a regular attention mask matrix in transformer encoder layer. The purpose of equation 11 could be explained from a perspective that a token in position $i$ only can interact with tokens in the same segment with it, and two particular tokens $[CLS]$ and $[seq]$. This way, we make the model concentrate on each segment in the following layer. And this mimics the process of humans doing CWS. According to formula 11, each token in a sentence could attend the hidden states of $[CLS]$ and $[seq]$. We think the model could easily achieve useful global context information and simultaneously reduce distraction from unnecessary information.

Then $M_{seg}$ and $H_L$ are input into the following transformer encoder layer. This transformer encoder layer is as same as the layer in the encoder module, except that its multi-head attention mask matrix is replaced by $M_{seg}$. We denote the output of this transformer encoder layer as $\hat{H}_L$ and we input $\hat{H}_l$ into a normal decoder and get sequential predictions as demonstrated in 4.3.

The whole loss of our proposed model is shown in equation 12. It is composed by normal decoder loss $L_D$ and boundary detection loss $L_{BD}$. Two scalars $\alpha$ and $\beta$ are used to balance these two losses.

$$L = \alpha L_D + \beta L_{BD} \tag{12}$$

## 5 EXPERIMENTS

We conduct an exhaustive set of experiments to examine the performance of BED-CWSM. We also do some ablation studies to prove the effectiveness of BED. As with other CWS-related studies, we also use Standard F1-score (F1) as evaluation metrics.

### 5.1 DATASETS

We size up our model on four benchmark CWS datasets, including PKU, MSR, AS, and CITYU. PKU and MSR are from SIGHAN 2005 Bakeoff Emerson (2005). AS and CITYU are in traditional Chinese characters, and we convert them into simplified ones as previous studies Chen et al. (2015); Tian et al. (2020) did. We follow the training/test split in WMseg Tian et al. (2020) for all four datasets. Table 1 shows the statistics of all datasets regarding the number of characters and words and the percentage of out-of-vocabulary (OOV) words in the dev/test sets concerning the training set.

### 5.2 EXPERIMENTAL SETTINGS

To examine the effectiveness and adaptability of our decoder. We employ our model with two different encoder architectures: BERT and WMSeg Tian et al. (2020). These two encoders rely on pre-trained BERT, and we use a pre-trained Chinese-BERT model, with 12 layers of transformer encoder, from huggingface[1]. We use WMSeg Tian et al. (2020) as our code base and modify its code to construct our model. We train WMSeg models ourselves, and all metrics of WMSeg reported are based on our models.

We also compare different regular decoders in our BED. We try softmax and crf decoders in our BED and size up their performance by combining them with two different encoders: BERT and WMSeg.

---

[1]https://github.com/huggingface/transformers

| Model | BED | PKU | | MSR | | AS | | CITYU | | Average | |
|---|---|---|---|---|---|---|---|---|---|---|---|
| | | F1 | OOV F1 | F1 | OOV F1 | F1 | OOV F1 | F1 | OOV F1 | F1 | OOV F1 |
| BERT+softmax | ✗ | 96.56 | 86.28 | 98.44 | 85.79 | **96.71** | 79.19 | 97.88 | 88.02 | 97.39 | 84.82 |
| | ✓ | **96.71**(+0.15) | **86.91**(+0.63) | **98.46**(+0.02) | **87.27**(+1.48) | 96.69(-0.02) | **79.6**(+0.41) | **97.91**(+0.03) | **88.25**(+0.23) | **97.44**(+0.05) | **85.51**(+0.69) |
| WMSeg+softmax | ✗ | 96.74 | 87.07 | 98.51 | 85.37 | **96.69** | 78.19 | 97.92 | **88.22** | 97.46 | 84.71 |
| | ✓ | **96.78**(+0.04) | **87.51**(+0.44) | **98.55**(+0.04) | **86.18**(+0.81) | 96.68(-0.01) | **79.04**(+0.85) | **97.95**(+0.03) | 87.85(-0.37) | **97.49**(+0.03) | **85.15**(+0.44) |
| BERT+crf | ✗ | 96.69 | 86.97 | 98.45 | 86.07 | 96.69 | 79.45 | 97.87 | **88.33** | 97.42 | 85.21 |
| | ✓ | **96.73**(+0.04) | **87.61**(+0.64) | **98.49**(+0.04) | **87.66**(+1.59) | **96.78**(+0.09) | **80.74**(+1.29) | **97.88**(+0.01) | 87.61(-0.28) | **97.47**(+0.05) | **85.32**(+0.11) |
| WMSeg+crf | ✗ | 96.75 | 87.04 | 98.52 | 86.28 | 96.75 | 79.71 | 97.88 | 87.44 | 97.47 | 85.12 |
| | ✓ | **96.76**(+0.01) | **87.05**(+0.01) | **98.56**(+0.04) | **86.61**(+0.33) | **96.79**(+0.04) | **79.98**(+0.27) | **98.04**(+1.84) | **88.35**(+1.11) | **97.53**(+0.05) | **85.50**(+0.18) |

Table 2: Performance of all models in the experiment. Models using BED outperform their baseline counterparts on average across all datasets. ✓indicates that the corresponding model uses BED, and a row marked with ✗means that the model uses a normal decoder. BERT and WMSeg represent which encoder is adopted in the model. "+softmax" and "+crf" represent the normal decoder type adopted in the model's decoder.

| Model | PKU | |
|---|---|---|
| | F1 | OOV F1 |
| BERT+softmax | 96.56 | 86.28 |
| +TFE | 96.67(+0.11) | 86.64(+0.36) |
| +BD | 96.67(+0.11) | **86.91**(+0.63) |
| +BD +TFE | 96.59(+0.03) | 86.61(+0.33) |
| +BED | **96.71**(+0.15) | **86.91**(+0.63) |

Table 3: Results of ablation study on PKU.

In our experiment, we set the threshold of boundary detection segmentation $thres_{seg}$ to 0.98. Two params of loss $\alpha$ and $\beta$, we only impose them to scale two sub-loss into the same order of magnitude. So we set $\alpha$ to 0.8 and $\beta$ to 1.

## 5.3 OVERALL RESULT

Table 2 shows results of all our experiments. It demonstrates that comparing with models with a normal decoder, models with BED achieve better performance, no matter what kind of encoder taken by them. The model with BERT encoder and softmax decoder using BED could improve its average F1 by 0.05% and average OOV F1 by 0.69%. For a model with a BERT encoder and crf decoder, BED could bring 0.05% benefits on average F1 and 0.11% benefits on average OOV F1. For a model that takes WMseg encoder, BED could boost the performance of the baseline model by 0.03% on average F1 and 0.44% on average OOV F1 when the model uses a softmax decoder, 0.05% on average F1 and 0.18% on average OOV F1 when it uses a crf decoder.

For datasets PKU and MSR, models with BED consistently outperform models without BED. For dataset AS, performances of BERT+softmax and WMSeg+softmax with BED are slightly worse than these models without BED. However, these models could achieve better performances on OOV F1 metric when selecting BED. For dataset CITYU, all models perform better when adopting BED on F1. The model which combines WMseg encoder and crf standard decoder always perform better on the four datasets when using BED. From experiment results, we also found that there is no one-size-fits-all system as Fu et al. (2020) did. For different datasets, models achieving the best performance do not have the same architecture. WMseg + softmax with BED achieves the best performance on PKU, and WMSeg + crf with BED achieves the best performance on the other three datasets.

## 5.4 ABLATION STUDY

To validate the effectiveness of the whole architecture of our proposed BED, we carry out a detailed ablation study on the PKU dataset. The results of the ablation study are shown in 3. In the ablation experiment, we use BERT + softmax as our baseline and evaluate the gains of two essential parts, transformer encoder layer(+TFE) and boundary detection module (+BD), in BED. And by comparing the model with a whole BED (+BED) and the model with a BED without a coarse-grained segmentation part (+BD +TEE), we try to prove the effectiveness of the coarse-grained segmentation mechanism.

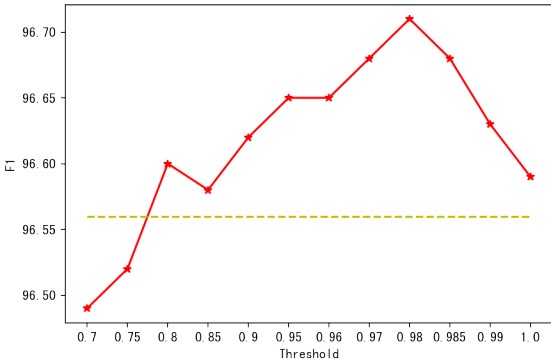

Figure 3: Performaces of models trained with different $thres_{seg}$ on PKU dataset.The yellow line is the perfomance of the baseline model.

Although the model adding TFE or BD could achieve better F1, 0.11% improvement compared with baseline, a model combining them straightforwardly without a coarse-grained segmentation mechanism does not get an effect of one plus one, only gets 0.03% improvement. That means combining TFE and BD is harmful to each other gains. A model with BED has a 0.15% improvement that reaches 96.71%, which is higher than the improvement attended by only using TFE or BD. When it comes to the performance of models on OOV words, the model with BED gains the same improvement as the model with BD, with 0.63% improvement over the baseline model that reaches 86.91%. So we could conclude that BED is an effective decoder structure.

## 5.5 EFFECT OF DIFFERENT THRESHOLD

We also study the effect of different values of $thres_{seg}$. We trained models which have our proposed BED with different $thres_{seg}$ and the results are shown in figure 3. We could notice that a lower or higher value of $thres_{seg}$ could decrease the performance of a model. We should select an appropriate $thres_{seg}$ value. However, when we choose a higher value, we always could get a model whose performance exceeds the baseline model.

## 5.6 CASE STUDY

To investigate how BED and corresponding coarse-grained segmentation part works, we select one example input sentence "全体/村民/过/半数/通过" (More than half of all villagers passed). In the baseline model without BED, it tokenizes "过半数"(more than half) as one word. The model with BED dose not find the boundary between "过" (more than) and "半数" (half) in the boundary detection module. However, the multi-grained decoder module in the model could split it correctly. Figure 4 shows the self-attention weights in BED of this example. We could see that "过半数" could only reach each other and two special symbols because of $M_{seg}$. And that makes the model could focus on information more related to one position and do fine-grained segmentation as humans do.

Another example is the sentence shown in Figure 1. We also visualize its corresponding boundary detection tags, self-attention weights and final segmentation tags in Figure 5 in appendix A. In the boundary detection module, the model splits "论证后金" as a whole. After multi-grained decoder module, the model tokenizes it into two words, "论证" and "后金". From the visualization, we could discover that the boundary detection module could find most of ending boundaries in the sentence and assign a high probability to the corresponding tag. This also proves that most of the words in the sentence are relatively easy to segment and we could tokenize them correctly in coarse-grained segmentation.

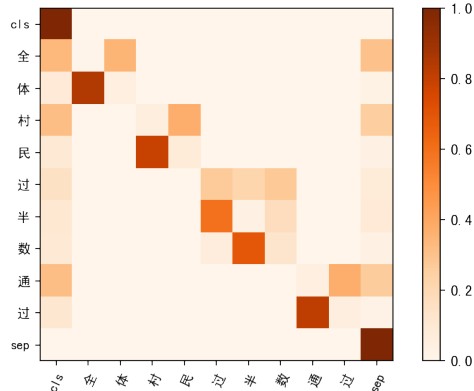

Figure 4: Heatmap of self-attention weights in the transformer encoder layer in BED.

## 6 CONCLUSION

In this paper, we proposed an effective decoder for CWS named Boundary-Enhanced Decoder (BED). To our best knowledge, it is the first study trying to optimize the decoder part in a CWS model. We conduct exhaustive experiments on varieties of encoders and typical decoders and prove the effectiveness of our proposed method. It could bring 0.05% and 0.69% improvement on Average-F1 and OOV Average-F1 on four benchmark datasets when using a BERT encoder and softmax standard decoder. And we also publish our source code of BED.

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

## A  APPENDIX

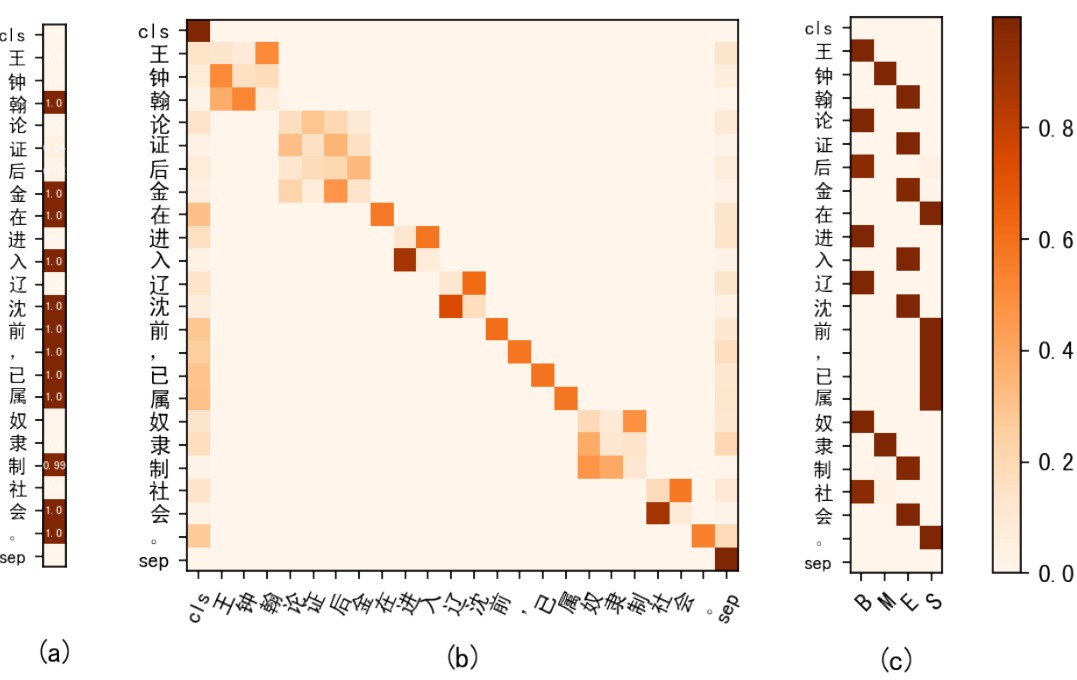

Figure 5: Visualization of boudary detection tags (a), attention weight of transformer encoder layer in BED (b), and final tags (c).

