# OpenReview forum: "BED: Boundary-Enhanced Decoder for Chinese Word Segmentation"
_ICLR.cc/2023/Conference — Submitted to ICLR 2023_

### Official Review · Reviewer_2k3q · 2022-10-23

**Confidence:** 4
**Correctness:** 3
**Technical Novelty And Significance:** 2
**Empirical Novelty And Significance:** 2
**Recommendation:** 5

**Clarity, Quality, Novelty And Reproducibility:**

- In Table 2, we can see accuracy get better by introducing the proposed approach, though do you have any comparison accuracy scores of the previous work?
- How did you create Chinese vocabulary inn your experiment? You should clarify it even if you use BERT model's vocabulary.
- The authors claim that the proposed approach addresses the OOV issue in Chinese, but I am not clear how to be resolved. Do you have a good example for this claim?

**Strength And Weaknesses:**

Strengths
- The idea is straightforward and relatively simple
- The results support the author's claim that the proposed approach works better

Weakness
- even though we see consistent improvement across dataset, the model's performance already looks saturated. From the human perspectives, e.g. +0.15 is significant difference? Did you do significant test against the baselines?

**Summary Of The Paper:**

Word segmentation is an essential preprocessing step in languages like Chinese that requires tokenization and boundary insertion. Most of studies in Chinese word segmentation focus on the encoder and do not refine the decoder part, to tokenize the segments into more fine-grained words. More accurately, the authors introduce another network called "Multi-grained decoder module" on top of the conventional CWS model architecutre. The authors tackles OOV problems in Chinese and experimentally show that the proposed approach of "Boundary-Enhanced Decoder" is effective in extensive experiments.

**Summary Of The Review:**

This paper proposes a simple and effective approach to improve Chinese word segmentation. By introducing a new network on top of the decoder, the authors addresses the OOV issue as well as words segmentation. The experimental results show that the proposed approach bring 0.05%-0.69% improvement on average F1 and OOV average F1 scores on multiple benchmark datasets. The authors only compare their results agains the vanilla Chinese word segmentation approaches and show the effectiveness, however, they also need to report the performance comparison agains the previous work.

---

### Official Review · Reviewer_xn6T · 2022-10-24

**Confidence:** 3
**Correctness:** 2
**Technical Novelty And Significance:** 2
**Empirical Novelty And Significance:** 1
**Recommendation:** 5

**Clarity, Quality, Novelty And Reproducibility:**

Some parts of this paper is unclear and needs further rework on improving the quality. In particular, an important part, i.e., the probability of span used for masking, is missing, that seems to be critical in the proposed model. Thus, this work is harder to reproduce.

**Strength And Weaknesses:**

Strength

* This work is interesting in that the task is decomposed into two steps, one for rough segmentation which is transformed as making for interpreting representations inputs with explicit boundaries before the final prediction.

* Small gains on top of conventional word embeddings, e.g., BERT.

Weakness

* It is a bit hard to follow the descriptions in this paper, and needs further work on improving presentation including mathematical notations.

* It is not clear how p is derived in Equation 11, which is probably representing a probability of a span. Note that p is used as a probability of segmentation in Equation 10 which is basically a binary decision, and i is used as a position and j is meant to represent the index to the training instance. However, the meaning for i and j seem to be different from previous equations.

* Given that the binary prediction is helpful in this work, it is not clear why this work still relies on BMES notation. Probably it would be good to report a model which simply do binary prediction.




**Summary Of The Paper:**

This work proposes an approach for Chinese word segmentation that is happening in the final prediction layer. First, a binary prediction is performed for each position as a preliminary judgement of word segmentation. Then, the binary decision is transformed as a span-wise decision representing a masking for self-attention mechanism. Finally, a CRF is used to assign tags, i.e., E, S, M, B, to perform word segmentation. Experiments are carried out on four standard Chinese word segmentation tasks and show consistent gains when using various vector representations, e.g., BERT.

**Summary Of The Review:**

This work presents an interesting work on Chinese word segmentation that combines binary decision to perform rough segmentation before the fine grained segmentation by CRF. However, this work has a clarity issue and needs further experiments to justify the results.

---

### Official Review · Reviewer_wqVM · 2022-10-25

**Confidence:** 3
**Correctness:** 3
**Technical Novelty And Significance:** 3
**Empirical Novelty And Significance:** 3
**Recommendation:** 5

**Clarity, Quality, Novelty And Reproducibility:**

- The boundary detection module is also related to transition-based approaches that iteratively perform the shift and append transitions to determine the boundaries of words. This paper doesn't follow board CWS approaches in related work.

- Possible Typo
Caption for Figure2:

Fine-grained Decoder Module -> Multi-grained Decoder Module ?


**Strength And Weaknesses:**

- The strength: authors achieve the better results with the proposed BED, although the improvements are relatively limited.

- The weakness is that it is questionable that the improvement of the decoder is novel. The related work is limited for the improvement of the decoder in CWS.


**Summary Of The Paper:**

This paper addresses the decoder improvements in Chinese Word Segmentation (CWS). The authors state the contribution of the previous CWS models are limited in the encoder.

They proposed the optimization of the decoder of Boundary-Enhanced Decoder (BED). Based on the conventional CRF decoder, the BED model introduces Boundary-Enhanced Decoder, which is composed of two modules: boundary detection module and fine-grained decoder module. Boundary detection module determines the end of the words. Multi-grained decoder module includes coarse-grained segmentation, a transformer encoder, and a normal decoder layer.

I do not consider that the improvement of the encoder part is novel for sequence labeling including CWS. In sequence labeling, many approaches are adopted including hierarchical softmax or refinement of attached labels (non-autoregressive). Considering these preceding studies, it is questioning, the improvement of the decoder is novel in CWS.


**Summary Of The Review:**

I'm not certain that the proposed model is novel and significant enough to reach the high-bar of the ICLR conference. I rather prefer this paper is also reviewed in a specific domain of the community.

---

### Official Review · Reviewer_1QHU · 2022-10-27

**Confidence:** 4
**Correctness:** 2
**Technical Novelty And Significance:** 2
**Empirical Novelty And Significance:** 2
**Recommendation:** 5

**Clarity, Quality, Novelty And Reproducibility:**

The innovation of this paper is not very strong. It aims to optimize decoder for the CWS model and announce that "it is the first study trying to optimize the decoder part in a CWS model". However, a lot of previous work on CWS indeed consider decode constraint and transition based CWS work could be viewed as a typical work managing decoder.

**Strength And Weaknesses:**

Strength:
1. The motivation of this paper is very clear to optimize decoder for the CWS model to better tackle OOV problem.
2. The paper is easy following.

Weaknesses:
1. The innovation of this paper is not very strong. It aims to optimize decoder for the CWS model and announce that "it is the first study trying to optimize the decoder part in a CWS model". However, a lot of previous work on CWS indeed consider decode constraint and transition based CWS work could be viewed as a typical work managing decoder.
2. A lot of related papers are also missing as point 1 states that some previous related work is not mentioned.
3. The experimental part is not very convincing. The comparison is not solid too. I didn't see previously work numbers well compared.




**Summary Of The Paper:**

CWS models are subjected to OOV words. This paper proposes an optimized decoder for the CWS model called Boundary-Enhanced Decoder (BED), which looks for easily tokenized positions, divides a whole sentence into several parts first, and then tokenizes each
piece into more fine-grained words. This more intuitive approach can further improve the performance of the word segmentation model.

**Summary Of The Review:**

I think this paper need to be revised and polished on both motivation part and experimental part.

---

### Decision · Program_Chairs · 2023-01-20

**Decision:**

Reject

**Justification For Why Not Higher Score:**

The proposed method does not make a clear theoretical or empirical contribution over existing approaches to Chinese Word Segmentation.

**Justification For Why Not Lower Score:**

N/A

**Metareview: Summary, Strengths And Weaknesses:**

The paper proposes a method that aims to improve the handling of out of vocabulary words for Chinese Word Segmentation by decomposing decoding into two steps. Results show a small improvement (around 0.1% F1) on standard benchmarks, but it is unclear how significant that is given the high performance of existing approaches on this task. Previous work has already investigated related multi-step approaches to determining word boundaries, so the novelty of the approach is limited and the paper does not make a clear enough contribution.